# Differences in Exercise Stress, Job Satisfaction, Intention to Quit Exercise, and Quality of Life According to the Psychological Abuse Experiences of Elite Male Athletes

**DOI:** 10.3390/bs14050392

**Published:** 2024-05-07

**Authors:** Mun-Gyu Jun, Soon-Young Kim

**Affiliations:** 1Department of Coaching, College of Physical Education, Kyung Hee University, Seocheon-dong 1, Giheung-gu, Yongin-si 17104, Republic of Korea; mkrollcage@khu.ac.kr; 2Department of Physical Education, Gachon University, 1342, Seongnam-daero, Sujeong-gu, Seongnam-si 13120, Republic of Korea

**Keywords:** psychological abuse, elite male athlete, exercise stress, job satisfaction, intention to quit exercise, quality of life

## Abstract

This study aimed to further understand psychological abuse in sports and contribute to the development of elite sports and athletes’ persistent performance by identifying the causal effects of psychological abuse on elite athletes’ exercise stress, job satisfaction, intention to quit exercise, and quality of life (QOL). Data were collected from 363 elite South Korean male athletes (ages ≥ 20 years) from August to September 2023. The independent variable for comparative analysis was the presence or absence of psychological abuse in elite male athletes by coaches. The participants were divided into two groups: a non-abuse-experienced group (Group 1) and an abuse-experienced group (Group 2). Participants’ demographic and athletic background information (e.g., career and sport) were also collected. This study showed that the three factors (exercise stress, intention to quit exercise, and QOL) were higher in Group 2 than in Group 1. These findings provide a meaningful analysis of the impact of psychological abuse on the mental health, persistence, and overall QOL of elite male athletes that can be used to develop countermeasures and policies against psychological abuse that threatens the mental health of elite athletes.

## 1. Introduction

Recently, coaches and senior staff physically abusing athletes for disciplinary purposes [1] has been recognized as a serious and ongoing problem in sports. South Korean society increasingly views sports organizations as “hotbeds of violence and corruption” and athletes as potential accomplices who have become “impure“ [2]. There are also concerns about the internalization and inheritance of a violent culture and moral hazards [3]. Consequently, violence in the sports world can be viewed as a permanent problem and a major challenge to overcome [4].

Surveys conducted by the National Human Rights Commission [3] and the Korean Sport and Olympic Committee [5] classify sports violence as sexual, physical, verbal, and emotional abuse, reporting that abuse causes bystander damage and has decision-making and economic effects. The phrase “invisible wounds can be deeper” [3] (p. 9) provides an overview of verbal abuse and the severe psychological and emotional impacts—i.e., the psychological distress [6]—of all forms of sports violence, including invisible violence. The International Olympic Committee considers psychological abuse a category of non-accidental violence, suggesting that it underlies the domains of physical and sexual abuse and negligence, wherein language is used as a mechanism for developing violence [7]. The psychological domain of sports violence has been recognized and should be included as a type of violence, along with physical and sexual types [8,9].

### 1.1. Exercise Stress

Exercise stress refers to feelings of threat and anxiety experienced by athletes when they encounter physically or psychologically difficult situations while participating in sports [10]. Moreover, worry about failure, the time required for training, financial burdens, doubts about talent, and interpersonal relationships cause stress for athletes participating in competitions [11]. Interpersonal relationships act as stress factors for South Korean athletes, which can be attributed to the hierarchical nature of the Korean culture [12,13,14]. The disparities between abilities and perceived demands can increase stress levels. Additionally, stress responses vary depending on the type of exercise or sport [15,16], and athletes experience greater stress in individual sports than in team sports because they are directly responsible for achieving their goals [15,16]. Regardless of the sport or cause of stress, it can damage athletic performance, lead to deviant behaviors, eating and sleeping disorders, and injuries, and reduce an individual’s athletic performance, job satisfaction, and quality of life (QOL) [17]. Consequently, it is critical to understand the causes of stress in athletes and develop methods to reduce it [18].

### 1.2. Job Satisfaction

When athletes are considered expert professionals, their job satisfaction is important and can take many forms [19]. For athletes, job satisfaction refers to an emotional attitude of interest, enthusiasm, and favor toward their job because their internal and external needs are satisfied during the job performance process [20]. It is also a psychological concept that refers to mental satisfaction and can generate emotional and affective responses [21]. Choi [22] argued that coaches should strive to fulfill the needs and satisfaction of athletes to enhance the atmosphere of sports teams and athletes’ morale. Choi [23] reported that athlete satisfaction refers to a response regarding the extent to which athletes’ needs are fulfilled, including satisfaction with athletic life, relationships, and fulfillment of athletic needs.

### 1.3. Intention to Quit Exercise

In sports, the intention to quit exercise refers to the cessation of continued participation in a sports program after a period of participation because of personal, social, or institutional factors [24]. Although sporting activities should be performed regularly and consistently to maximize the benefits of exercise, many people drop out within a few months of starting them [25]. Motivational factors, such as not being as good as expected, wanting to play another sport, disliking violence, boredom, dislike for the coach, excessively difficult training, and not being stimulating enough, are the major psychological factors influencing the intention to quit exercise [26,27]. The intention to quit exercise is commonly interchangeable with dropping out, retiring, or abandoning exercise. However, because terms such as dropping out and abandonment have negative connotations, this study uses intention to quit exercise instead [28]. It overcomes the limitations of these terms and encompasses the multidimensional factors that affect athletes’ intentions to quit exercise. Furthermore, including the “intention to quit exercise” enables a broader and new direction in athlete discussions, aligning with the recent paradigm shift in how athletes are viewed [29].

### 1.4. Quality of Life

Humans strive continuously for happiness. Happiness is not proportional to wealth or living conditions. Both objective standards (i.e., wealth and living conditions) and subjective evaluations contribute to human happiness. Human happiness and QOL have become common topics of study in many academic disciplines [30]. QOL, as defined by the World Health Organization, is influenced by culture, education, customs, and other factors in a region’s community [31]. Individuals living in a community have different goals, interests, expectations, standards, and so on [31]. QOL comes from recognizing things relevant to one’s social position or role [31]. It refers to the overall satisfaction with one’s standard of living. Specific subjective QOL evaluation items include economic level, psychological satisfaction, overall satisfaction with life, and social welfare levels [32], particularly happiness and QOL satisfaction [33]. Life satisfaction, expressed as happiness or contentment or a subjective feeling or mindset, is the most important component of mental health. Life satisfaction is also essential for individual and collective well-being because it affects individuals, their families, and the larger society [34]. QOL is subjectively perceived and can be understood through one’s life experiences [35], primarily referring to an individual’s feelings of happiness [36].

### 1.5. Research Purpose and Hypotheses

The conceptualization and construction of psychological abuse in the context of sports violence in South Korea have not been thoroughly assessed. A framework is required to understand violence that has been hidden, unrecognized, or trivialized, and a conceptualization of this invisible violence is required to better understand violence in sports. Therefore, this study aimed to explore psychological abuse in sports and help in the development of elite sports along with athletes’ continued activities by identifying the causal effects of psychological abuse on athletes’ exercise stress, job satisfaction, intention to quit exercise, and QOL. The four research hypotheses established in this study were:

**Hypothesis** **1.**
*Elite male athletes with experience of psychological abuse will have a higher level of exercise stress than those without experience of psychological abuse.*


**Hypothesis** **2.**
*Elite male athletes with experience of psychological abuse will have a lower level of job satisfaction than those without experience of psychological abuse.*


**Hypothesis** **3.**
*Elite male athletes with experience of psychological abuse will have a higher level of intention to quit exercise than those without experience of psychological abuse.*


**Hypothesis** **4.**
*Elite male athletes with experience of psychological abuse will have a lower level of quality of life than those without experience of psychological abuse.*


## 2. Materials and Methods

### 2.1. Survey Participants and Data Collection Procedure

This study investigated the differences in exercise stress, job satisfaction, exercise cessation, and QOL according to the psychological abuse experiences of elite male athletes to determine the effects of psychological abuse experiences on their mental health and life. The study participants were elite male athletes aged >20 years from the Republic of Korea. The term elite athlete used in this study refers to an individual who is registered with the Korea Sports Federation and has chosen athletics as a career as well as a student–athlete. In addition, the study excluded minors under the age of 20 years, a vulnerable population. Data were collected from August to September 2023. This study used a quantitative research design and convenience and purposive sampling. Google Forms, an online survey platform, was used for the data collection. Offline survey questionnaires were used according to the participants’ choices. Data were collected with permission from the athletic departments and head coaches of three universities in a metropolitan area of the Republic of Korea. All participants voluntarily participated in the survey using the self-administration method and were informed of the purpose of the study. A total of 363 questionnaire responses were collected.

Participants’ essential demographic information (e.g., sex and age) was collected, along with additional information (i.e., athletic career). The athletic career was based on the survey of participants enrolled in the Korean Olympic Committee as elite athletes. Based on the participants’ answers, they were divided into two groups according to their psychological abuse experiences during their athletic careers (Group 1: psychological non-abuse-experienced group and Group 2: psychological abuse-experienced group). Psychological abuse experience was set as an important independent variable for this comparative analysis. In addition, participants answered follow-up questions regarding (a) persons who perpetrated psychological abuse and (b) the presumed reasons for psychological abuse. The demographic information of the participants is presented in Table 1. Finally, to compare and analyze the mental health of elite athletes, the main survey collected data on four factors (38 items in total): (a) exercise stress, (b) job satisfaction, (c) exercise cessation, and (d) QOL.

### 2.2. Instruments

First, instruments were used to measure male athletes’ exercise stress and intention to quit exercise based on their psychological abuse experience by modifying the factors used in a previous study [11] that analyzed the effects of exercise stress on continuous exercise and intention to quit exercise among professional athletes. The exercise stress factor consisted of four sub-factors: (a) coaching (five items), (b) career (five items), (c) performance (four items), and (d) privacy (four items). In addition, the intention to quit exercise had two sub-factors: (a) an internal factor (four items) and (b) an external factor (four items). Next, we applied an instrument modified by Choi [22] to measure job satisfaction factors in order to investigate elite athletes’ job commitment and satisfaction. This tool consisted of two factors: (a) intrinsic (four items) and (b) extrinsic (four items). Finally, the instrument used to analyze the QOL as an elite athlete was applied using Kim’s tool [27] to study professional soccer players’ career decisions, self-efficacy, mental well-being, and QOL. The QOL measurement tool consisted of four items as a single factor. All survey items applied in this analysis were rated on a five-point Likert scale ranging from 1 (“not at all”) to 5 (“very much”).

### 2.3. Data Analysis

The data collected in this study were analyzed using SPSS version 23.0. First, the analysis generated descriptive statistics for the study participants, including sociodemographic information (e.g., age, athletic career, perpetrator of psychological abuse, and reasons for psychological abuse). Second, this study tested the validity of the collected data using confirmatory factor analysis (CFA) with four dependent variables (exercise stress, job satisfaction, exercise cessation, and QOL). Third, to verify the scale reliability of the data, Cronbach’s alpha coefficients were tested for each factor. Finally, multivariate analysis of variance (MANOVA) was performed to determine the differences in all factors between Groups 1 and 2. MANOVA is a reliable statistical technique that has been widely applied in social science research [37,38,39,40,41,42] for comparative study designs.

## 3. Results

### 3.1. Scale Validity and Reliability

First, CFA was implemented to test the scale validity using nine dependent variables, including the sub-factors. Consequently, CFA secured the statistical standards. The goodness-of-fit test revealed that the statistical criteria were satisfied (CMIN = 1111.893; DF = 629; CMIN/DF = 1.768; NFI = 0.833; CFI = 0.919; and RMSEA = 0.055). Second, Cronbach’s alpha coefficients were tested to investigate the reliability of the survey questionnaires on each factor based on a statistical cutoff value of 0.70 [43]: coaching (exercise stress), *α* = 0.899; career (exercise stress), *α* = 0.933; performance (exercise stress), *α* = 0.918; privacy (exercise stress), *α* = 0.899; intrinsic job satisfaction, *α* = 0.824; extrinsic job satisfaction, *α* = 0.824; internal factor (intention to quit exercise), *α* = 0.842; external factor (intention to quit exercise), *α* = 0.782; and QOL, *α* = 0.820. Therefore, all variables applied in this study had satisfactory internal statistical reliability. More detailed results on the scales’ validity and reliability are presented in Table 2.

### 3.2. Multivariate Analysis of Variance

MANOVA was conducted to verify the differences in the dependent variables (exercise stress, job satisfaction, intention to quit exercise, and QOL) (Table 3). First, the homogeneity of covariance was secured (Box’s *M* = 115.668, *F* = 2.487, *p* < 0.001). Statistically significant differences were found between the two groups (Wilks’ Lambda = 0.715, F = 15.603, *p* < 0.001, partial *η^2^* = 0.285). The comparative analysis revealed statistically significant mean differences in coaching (exercise stress), career (exercise stress), performance (exercise stress), privacy (exercise stress), internal factor (intention to quit exercise), external factor (intention to quit exercise), and QOL. Group 2 reported higher mean scores than Group 1 for the seven factors. The detailed mean scores of all dependent variables in the two groups are shown in Table 4.

## 4. Discussion

This study examined psychological violence as a key factor in Korean elite male athletes and compared its effects on exercise stress, job satisfaction, intention to quit exercise, and QOL. Psychological abuse by coaches can have a significant impact on elite athletes’ careers [44]. Psychological abuse, which often destroys athletes’ lives, has received less attention than physical abuse. For elite athletes who are expected to perform at their peak in the face of fierce competition, the experience of psychological abuse could be crucial to their lives, both as athletes and as individuals. Furthermore, as elite athletes have a relatively early retirement age compared to other professions, psychological abuse that hinders elite athletes from performing at their best must be eradicated. This research topic provides essential objective data for improving the lives and mental health of elite athletes. The significant findings of this study are outlined below.

First, the exercise stress factor, consisting of four sub-factors (career, coaching, privacy, and performance), showed relatively high results across all sub-factors in the group of elite male athletes who had experienced psychological abuse. The first factor, career, reflects concerns about elite athletes’ relatively short professional lifespan. Notably, these concerns were higher in those who experienced psychological abuse. While each elite athlete’s perception of satisfactory performance may be subjective, performance is key to their future career path [45]. Psychological abuse can be caused by a lack of satisfactory performance in an atmosphere in which only a few elite athletes are recognized for their outstanding performance. Most elite athletes may have had such experiences, especially during adolescence, when they may have focused on their athletic performance rather than academic performance [46]. Furthermore, athletes who are psychologically abused by coaches may experience increased anxiety and stress [47].

Second, the coaching stress factor also showed higher results in the group with psychological abuse, which is closely related to direct psychological abuse. This suggests that these athletes have experienced psychological abuse not only in the past but also in the present as adults. Unsurprisingly, the results were higher for those who had experienced psychological abuse than for those who had not. It is interesting to note that these elite male athletes have experienced psychological abuse as adults rather than as minors and that their coaches may perceive these violent methods as effective coaching methods [48]. Given the need for athletes to excel in intense competitions, a more scientific and objective training approach is required.

Third, a significant finding of this study was the performance stress factor, based on the finding that mean scores were notably higher in the group with psychological violence abuse. This indicates that participants who have experienced psychological abuse are more likely to be dissatisfied with their performance, implying that their objective performance may be lower than that of elite male athletes who have not experienced psychological abuse. Although stress is subjective to individuals, it can be inferred from the statistically significant results. Finally, the privacy stress factor was higher in the group that had experienced psychological abuse, and the survey responses suggested that coaches controlled the personal privacy of elite athletes. This could comprise a serious privacy violation because elite athletes are not minors.

The stress factors in this study included four sub-factors that appeared to be closely related to each other. The results showed that respondents who experienced psychological abuse had higher scores for all factors. The interpretation of all four factors together could indicate that elite male athletes who have experienced psychological abuse are likely to be insecure about their athletic careers because of their low levels of performance. Moreover, coaches can use psychologically abusive training methods to improve athletes’ performance and have the potential to control athletes’ personal lives. The psychological stressors of being an elite athlete and surviving in the competitive world are real [48]. However, the most noteworthy aspect is the use of psychological abuse as a means of achieving competitive performance. Psychological abuse should be considered violence, such as physical abuse [49].

Another factor that showed a statistically significant group difference in this study was the intention to quit exercise. Previous studies [50,51] have shown that elite athletes tend to have shorter lifespans than those in the general workforce and may have difficulty transitioning from elite athletics to other careers. Therefore, the decision to quit should be seriously considered. Two types of factors contribute to the decision to quit: internal, which refers to the athlete’s own perceived need to quit, and external, which is influenced by economic factors or relationships with others (coaches, family, and peers) [11,24]. The results of this study showed that elite male athletes who had experienced psychological abuse had higher scores for both factors. This is consistent with the aforementioned results for exercise-related stress factors. Athletes who had experienced psychological abuse were more likely to be stressed about their performance and anxious about an unclear career path. This may be related to internal factors that lead to athletes’ intentions to quit exercise, as they are worried about their performance. In addition, the external factor of the intention to quit exercise was statistically higher in the group that had experienced psychological abuse, where the stress caused by conflict with leaders was high. This suggests that psychological pressure is not related to the athlete’s perceived performance but rather to relationships with others who are important to the continuation of an elite athlete’s life. This suggests that psychological abuse, stress, and intention to quit exercise are all closely related.

The final factor in this study, QOL, showed results requiring further interpretation. Specifically, elite male athletes who experienced psychological abuse were found to experience relatively higher levels of QOL than those who had no experience of psychological abuse despite having high levels of exercise stressors (performance, career, privacy, and coaching) and intentions to quit exercise. This result was contrary to what was expected when the study was designed, and interpreting the results requires a discussion of factors that were not accounted for in this study. Elite athletes are expected to perform consistently at the top of their game. This competitive performance comes with physical and psychological sacrifices but is acceptable if the outcome is satisfactory [52]. In this respect, despite the psychological abuse, stress, and mental pressure to stop being competitive the elite athletes in this study had experienced, these sacrifices likely allowed for high levels of performance, and their subsequent satisfaction with having a competitive performance ensured QOL. Eventually, these findings will need to be supplemented with more objective data, such as objective performance levels and results of elite athletes. In addition, there is a need to confirm the correlation between psychological sacrifice and satisfactory performance and to determine which value should be prioritized. Furthermore, the values prioritized by each elite athlete may differ, requiring a multifaceted analysis.

## 5. Conclusions and Limitations

This study compared and analyzed the effects of psychological abuse on elite athletes, focusing on differences in exercise stress, QOL, job satisfaction, and intention to quit exercise. In particular, this study segmented survey participants into a non-abuse-experienced group (Group 1) and an abuse-experienced group (Group 2) and conducted comparative analyses.

The results of this study showed that Group 2 evaluated exercise stress and intention to discontinue exercise more negatively than Group 1, whereas QOL was positively evaluated. The results of this study are meaningful because they indicate that the experience of psychological abuse can significantly affect exercise stress, QOL, job satisfaction, and the intention to quit exercise among elite athletes. However, few studies have examined the psychological and emotional stability and healing of victims of psychological abuse. Therefore, the following suggestions are made based on the results of this study. First, this study examined only psychological abuse among elite athletes; further studies are needed to determine the factors influencing physical and sexual abuse. Second, it is necessary to conduct repeated longitudinal and multidimensional studies that consider the in-depth variables influencing psychological abuse to compensate for the limitations of studies that use self-reported questionnaires to measure variables. Third, social and institutional mechanisms are required to reduce psychological abuse using the perpetration and victimization factors identified in this study. Fourth, it is necessary to develop and validate effective intervention programs to manage the damage caused by psychological abuse and its perpetrators. The perspective of psychological abuse in sports must be broadened, and victims’ suffering must be acknowledged. Therefore, to address the underlying issues of psychological abuse, systematic programs should be developed, awareness should be raised among sports community members, and a foundation should be laid for appropriate interventions and counseling for athletes suffering from trauma. Fifth, experience of psychological abuse was a major factor in subdividing survey participants, but responses that relied on the survey respondents’ past subjective memories were bound to be the baseline. The self-reporting tools for psychological abuse, Violence to Athlete Question (VTAQ) and Interpersonal Violence Gaines in Sport (IVIS), were used for psychological abuse studies [53,54,55], but both were developed for younger athletes (ages 14–17). Therefore, more objective criteria for psychological abuse in adult athletes need to be developed. Last, in this study, there was a difference (uneven sample size) in the size of the two groups according to the experience of psychological violence. To achieve a statistically appropriate design, the limitations of realistic data collection must be addressed.

In recent years, physical abuse has been recognized as a serious criminal act in South Korea, and intervention measures for perpetrators and victims have been evaluated from various perspectives. However, psychological abuse is still considered a common form of interpersonal conflict; consequently, its negative impact on victims has been ignored. The goal of studying psychological abuse is to help people achieve sound mental health and psychological well-being beyond the absence of physical abuse. Therefore, psychological abuse must be recognized as a form of abuse, awareness of its effects should be raised, and concrete solutions to deal with it should be established at the earliest. Even in sports, athletes who have experienced psychological abuse can experience negative psychological crises, which may result in their intention to quit exercising. This sense of crisis may traumatize athletes and impede their maturation.

## Figures and Tables

**Table 1 behavsci-14-00392-t001:** Descriptive statistics.

Participant Characteristics	Subcategories	Group 1 Non-Abuse-Experienced (*n* = 251)	Group 2 Abuse-Experienced (*n* = 112)
Age	20 s	113 (45.0%)	22 (19.6%)
	30 s	103 (41.0%)	61 (54.5%)
	40 s	25 (10.0%)	23 (20.5%)
	Over 50	10 (4.0%)	6 (5.4%)
Type of sport	Individual	217 (86.5%)	96 (85.7%)
	Team	34 (13.5%)	16 (14.3%)
Athletic career (years)	Less than 1 year	-	-
	1–less than 5 years	168 (66.9%)	52 (46.4%)
	5–less than 10 years	56 (22.3%)	41 (36.6%)
	10–less than 15 years	18 (7.2%)	16 (14.3%)
	Over 15 years	9 (3.6%)	3 (2.7%)
Person who perpetrated the psychological abuse	Coach	-	55 (49.1%)
	Teammate	-	56 (50.0%)
	Family	-	-
	Others	-	1 (0.9%)
Reasons for the psychological abuse	Mistake during exercise	-	56 (50.0%)
	Mistake during game	-	16 (14.3%)
	Low performance	-	25 (22.3%)
	Personal mistake	-	14 (12.5%)
	Other	-	1 (0.9%)

**Table 2 behavsci-14-00392-t002:** Results of CFA (validity) and Cronbach’s alpha (reliability).

Construct and Scale Items	*λ*	AVE	C.R.	*A*
**Exercise stress**				
**Coaching**				
I am stressed because the exercises are boring or forced.	0.775			
I am stressed because my coach is violent during practice.	0.781			
I am stressed because I do not see eye to eye with the coach.	0.781			
I am stressed out when coaches scold me for no reason.	0.806	0.629	0.894	0.899
I am stressed because I have exercised a lot.	0.820			
**Career**				
I am stressed because I do not think I will make the professional team.	0.765			
I am stressed out when I do not get to play.	0.841			
I am stressed by the pressure of competition results.	0.845			
I am stressed about career decisions.	0.812	0.681	0.914	0.933
I am stressed because I do not think I can succeed.	0.860			
**Performance**				
I am stressed because my workouts do not go my way.	0.766			
I am stressed when bouts do not go well.	0.836			
I am stressed because the game is bad.	0.896	0.730	0.915	0.918
I am stressed because I do not perform as well as during practice.	0.911			
**Privacy**				
I am stressed when I do not get a break.	0.815			
I am stressed because I do not get to enjoy my free time.	0.803			
I am stressed when I work out, even on holidays.	0.835	0.659	0.885	0.899
I am stressed when my privacy is violated.	0.793			
**Job satisfaction**				
**Intrinsic satisfaction**				
I am satisfied with my sense of accomplishment as an athlete.	0.684			
I am satisfied to be recognized by others as an athlete.	0.785			
I decide everything for myself as an athlete.	0.694	0.509	0.805	0.824
I am satisfied with the rewards as an athlete.	0.685			
**Extrinsic satisfaction**				
I am satisfied with people’s interest in athletes.	0.700			
I am satisfied with my practice.	0.733			
I am satisfied with my social status as an athlete.	0.732	0.522	0.814	0.824
I am satisfied with my current conditions and training environment.	0.725			
**Intention to quit exercise**				
**Internal factor**				
I thought about quitting because the workouts were too hard.	0.549			
I thought about quitting the workout because I was not seeing any benefits.	0.810			
I thought about quitting because I am not as athletic as my peers.	0.821	0.560	0.833	0.842
I do not plan to continue working out in the future.	0.781			
**External factor**				
I thought about quitting working out because of a conflict with my peers.	0.521			
I thought about quitting sports because of my parents’ disapproval.	0.826			
I thought about quitting working out because of the cost.	0.738	0.501	0.797	0.782
I thought about quitting because of a conflict with a manager or coach.	0.712			
**Quality of life**				
I have pride and confidence in myself.	0.812			
I try to accomplish the important things I want in my life.	0.675			
I am satisfied with my life.	0.750	0.554	0.832	0.820
My life situation is generally good.	0.735			
CMIN = 1111.893, DF = 629, CMIN/DF = 1.768, NFI = 0.833, CFI = 0.919, RMSEA = 0.055				

Note. *λ* = Factor loading value; AVE = Average variance extracted; C.R. = Composite reliability; *α* = Cronbach’s alpha; CMIN = Chi-square; DF = Degree of freedom; NFI = Normed fit index; CFI = Comparative fit index; RMSEA = Root mean square error of approximation.

**Table 3 behavsci-14-00392-t003:** Results of multivariate analysis of variance.

Variables	Sub-Factors	*df*	*F*	*p*	*η* ^2^
Exercise stress	Career	1	36.925	0.000 ***	0.093
	Coaching	1	80.281	0.000 ***	0.182
	Privacy	1	26.486	0.000 ***	0.068
	Performance	1	62.144	0.000 ***	0.147
Job satisfaction	Intrinsic	1	0.258	0.612	0.001
	Extrinsic	1	0.268	0.605	0.001
Intention to quit exercise	Internal factor	1	10.187	0.002 **	0.027
	External factor	1	28.119	0.000 ***	0.072
Quality of life		1	7.755	0.006 **	0.021

Note. *** *p* < 0.001, ** *p* < 0.01.

**Table 4 behavsci-14-00392-t004:** Mean scores of the nine dependent variables on each group.

	Exercise Stress				Job Satisfaction		Intention to Quit		Quality of Life
	Career	Coaching	Privacy	Performance	Intrinsic	Extrinsic	Internal	External	
G1	2.32	2.22	2.42	2.50	3.28	3.23	2.78	2.35	3.50
G2	**3.06**	**3.13**	**3.02**	**3.97**	3.32	3.27	**3.11**	**2.92**	**3.72**

Note. Group 1 = psychological non-abuse-experienced group; Group 2 = psychological abuse-experienced group. Statistically significant higher mean scores among the groups are shown in bold.

## Data Availability

The data reported in this research are available on request from the corresponding author.

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
