# Peer review of "Differences in Exercise Stress, Job Satisfaction, Intention to Quit Exercise, and Quality of Life According to the Psychological Abuse Experiences of Elite Male Athletes"

_behavsci, 2024, doi:10.3390/bs14050392_

Round 1
Reviewer 1 Report
Comments and Suggestions for Authors
Dear authors,
Thank you very much for your contribution. Here are some questions that may be useful to improve the quality and understanding of the article:
ABSTRACT
- In the abstract it is stated that: The independent variable for the comparative analysis was the presence/absence of psychological abuse of elite male athletes by coaches. After reading the paper, it seems to me that maybe “by coaches” must be removed.
INTRODUCTION
- Review text lines 38-39: bystander damage twice.
- Try to avoid using quotation marks for specific terms; it is advisable to reserve them for textual quotations from the reference text, always giving the page number where it appears.
- The research objectives should be listed at the end of the introduction section, before Materials and Methods, rather than where they are currently (lines 47-54).
- Line 113; remove extra point.
MATERIALS AND METHOD
- It would be useful, within Materials and Methods, to detail the requirements to be considered as an elite athlete, as this may differ from country to country and needs to be explained for possible comparative purposes. Similarly, why have only male athletes been included in this study, and why only those over the age of 20? No information about these decisions is provided in the paper and I believe it may be relevant.
- It would be useful to detail if there is approval by an Ethics Committee, as well as the institution and code.
- It is detailed that the sample was selected by convenience and purposive sampling. How was this carried out? Was collaboration requested from any institution? How was the study disseminated in order to recruit participants?
- In the abstract it is stated that participants were divided into two groups: an abuse-experienced group (Group 1) and an abuse-non-experienced group (Group 2) but, in Materials and Methods it’s the opposite. Could you please check the whole document?
- It would be interesting to know, if you have the information, the type of sport and whether it was individual or team sport.
- It would be appropriate to change the title of Table 1, as it presents the socio-demographic information of the participants, so descriptive statistics can be misleading.
- What specifically does athletic career refer to? They are elite athletes and, although it would depend on the type of sport, the vast majority of them detail a duration of between 1 and 5 years. If in this case it refers to their start in sport, it would seem to me to be scarce, so perhaps it would be necessary to detail if it refers to the start of their participation in high-level competition.
- In Table 1, to avoid confusion regarding the last row where the total is detailed, as it is not a sum but the sample size of each of the groups, I would eliminate it. Instead, I would detail n= group size in the top cell where appropriate.
DISCUSSION
- The information contained in the first paragraph of the discussion should be substantiated by citing research that supports these findings. The same for lines 225-227. The same for lines 264-267.
- As with previous results…The scientific articles referred to must be referenced.
- In general, as far as the discussion is concerned, I miss the comparison of the results with more previous benchmark research, which would help to understand the findings of the research.
CONCLUSIONS AND LIMITATIONS
- As a reminder, please review the characteristics of group 1 and 2 throughout the document, as the profiles are still interspersed and may lead to doubt in the interpretation of results.
Author Response
I have attached a file (author's note).

Reviewer 2 Report
Comments and Suggestions for Authors
Thank you for letting me review your paper. The paper focuses on the less-studied, but evenly important factor of abuse, i.e. psychological violence. By using a quantitative design, it is also possible to examine the effects of experiencing psychological violence on athlete's mental health in more detail and on a broader scale. However, there is some information missing in the method section of the paper, which can hamper the interpretation and importance of the results and discussion. In the document separately uploaded, you can find a more detailed overview.

Below are some small suggestions for possibly rephrasing some sentences or words.
L17: I would use ‘non-abuse-experienced group’
L134: change ‘emotional abuse’ to ‘psychological abuse’
L87 ‘not exciting enough’ à I think this is a spelling mistake? Do you mean the sport not being exciting enough or the athlete not being excited enough?
L168: no need to repeat that analyses are performed via SPSS version 23.0.
L206 + 211 + 225-227: this sentence needs changes, they are not easy to read and understand at the moment.
L299: could you reword this, or what do you mean with ‘crime’ ?
Author Response
I have attached a file (author's note).

Reviewer 3 Report
Comments and Suggestions for Authors
Please see the attached file.

Author Response
I have attached a file (author's note).

Round 2
Reviewer 1 Report
Comments and Suggestions for Authors
Dear authors,
Having reviewed the suggested changes, I believe that the manuscript can be considered for publication.
Kind regards.
Author Response
Dear Reviewer,
Thank you for your support.
Reviewer 2 Report
Comments and Suggestions for Authors
Thank you for answering my questions and making some changes in the manuscript, this alreay improved the quality and clearity of the manuscript. However, there still remain some questions, which can be found in the separate word file.

Thank you for rewording or rephrasing several sentences. It would be good to reread the full manuscript once revised, since there are still some small errors in the text (e.g., repetition of a sentence at L257-260, or repitition word 'prevention' at L216-217).
Author Response
Dear Reviewer,
Thank you for you support.
Please check the attached file.

Reviewer 3 Report
Comments and Suggestions for Authors
I think that the changes you did during the review process were good, and that your scientific article is much better now (and is suitable for publishing into the journal).
I suggest you to reconsider (during the proofing process) the following small consideration:
Line 146 – It is written “20 items in total“, but from the lines 154 – 159 I have counted 38 items at all. Please, just check the line 146.
Author Response

(The authors gave the same response as above.)

Round 3
Reviewer 2 Report
Comments and Suggestions for Authors
Thank you for clarifying some of the remaining issues. I congratulate the authors with the revised manuscript, very interesting topic!